# Long-term memory CD8+ T cells specific for SARS-CoV-2 in individuals who received the BNT162b2 mRNA vaccine

Nozomi Kuse[1,3], Yu Zhang [1,3], Takayuki Chikata[1,3], Hung The Nguyen[1], Shinichi Oka[1,2], Hiroyuki Gatanaga[1,2] & Masafumi Takiguchi [1] ✉

Long-term memory T cells have not been well analyzed in individuals vaccinated with a COVID-19 vaccine although analysis of these T cells is necessary to evaluate vaccine efficacy. Here, investigate HLA-A*24:02-restricted CD8+ T cells specific for SARS-CoV-2-derived spike (S) epitopes in individuals immunized with the BNT162b2 mRNA vaccine. T cells specific for the S-QI9 and S-NF9 immunodominant epitopes have higher ability to recognize epitopes than other epitope-specific T cell populations. This higher recognition of S-QI9-specific T cells is due to the high stability of the S-QI9 peptide for HLA-A*24:02, whereas that of S-NF9-specific T cells results from the high affinity of T cell receptor. T cells specific for S-QI9 and S-NF9 are detectable >30 weeks after the second vaccination, indicating that the vaccine induces long-term memory T cells specific for these epitopes. Because the S-QI9 epitope is highly conserved among SARS-CoV-2 variants, S-QI9-specific T cells may help prevent infection with SARS-CoV-2 variants.

SARS-CoV-2 mRNA vaccines have been shown to prevent SARS-CoV-2 infection and reduce hospitalization and mortality rates[1–7]. These vaccines effectively elicit the production of both antibodies and T cells against SARS-CoV-2 spike antigens[8–18]. However, several countries have experienced fifth or sixth pandemic waves even when >70% of the population was vaccinated with two or three doses of SARS-CoV-2 mRNA vaccines. Increased infection and hospitalization rates have also been observed for vaccinated individuals. These increases likely stem from waning vaccine immunity over time. Indeed, it has been reported that antibody titers decline 4–6 months after vaccination[19–21]. The effectiveness of antibodies against the Delta and Omicron variants was also reduced[22–29], whereas a third dose of mRNA vaccine-elicited neutralizing antibodies against these variants[30,31].

Both antibodies and T cells are key immune mediators involved in preventing SARS-CoV-2 infection and disease onset. The production of antibodies against SARS-CoV-2 antigens is well documented in SARS-CoV-2-infected or vaccinated individuals[32,33], while there are also accumulating data on SARS-CoV-2-specifc T cells. Recent studies demonstrated that SARS-CoV-2-specifc T cells were elicited during the early phase of infection[16,17,34–38], and spike antigen-specific T cells were generated 1 week after the first vaccine dose[38,39]. Although many studies reported a large number of T cell epitopes (https://www.iedb.org), these studies identified T cell epitopes from predicted epitope peptides or overlapping peptides using various screening methodologies such as ELISpot, ICS, AIM, and multimer staining assays[40]. Therefore, some of these reported epitopes may not be optimal epitopes and their HLA-restriction has not been confirmed. Confirmation and characterization of these reported epitopes and identification of immunodominant SARS-CoV-2 epitopes are necessary for studies of SARS-CoV-2-specifc T cells and the development of efficient COVID-19 vaccines.

It remains controversial as to how long effective SARS-CoV-2 immunity persists in vaccinated individuals. Immunity against SARS-CoV-2 has been well evaluated based on the levels of antibodies against SARS-CoV-2 spike antigens or neutralizing antibodies in vaccinated individuals. Recent studies showed that COVID-19 vaccine effectively

[1]Division of International Collaboration Research and Tokyo laboratory, Joint Research Center for Human Retrovirus Infection, Kumamoto University, Kumamoto/Tokyo, Japan. [2]AIDS Clinical Center, National Center for Global Health and Medicine, Tokyo, Japan. [3]These authors contributed equally: Nozomi Kuse, Yu Zhang, Takayuki Chikata. ✉e-mail: masafumi@kumamoto-u.ac.jp

induced spike-specific effector CD8[+] T cells[41–44]. Because memory CD8[+] T cells may contribute to the long-term effects of the vaccine, analysis of this T cell population is critical to evaluate vaccine efficacy and determine the need for booster vaccinations. However, long-term memory T cells in the vaccinated individuals have been analyzed in a limited number of studies[21,45,46].

In the present study, we investigate CD8[+] T cells specific for SARS-CoV-2 spike epitopes in individuals administered the BNT162b2 mRNA vaccine and the effects of the mRNA vaccine on the maintenance of long-term memory T cells. We focus on CD8[+] T cells specific for spike epitopes restricted by HLA-A*24:02, which is a common HLA allele worldwide, particularly in Asian populations[47,48]. The results of this study show that long-term memory T cells specific for immunodominant and conserved epitope are elicited in individuals who received the BNT162b2 mRNA vaccine. These memory T cells are expected to prevent infection with SARS-CoV-2 variants and to suppress the replication of these viruses.

## Results

### HLA-A*24:02-restricted CD8[+] T cells elicited in individuals who received the BNT162b2 mRNA vaccine

To investigate the ability of COVID-19 mRNA vaccination to elicit SARS-CoV-2-specific CD8[+] T cells and maintain epitope-specific memory T cells, we analyzed HLA-A*24:02-restricted spike epitope-specific CD8[+] T cells in individuals who received two doses of the Pfizer-BioNTech BNT162b2 mRNA vaccine. We recruited 17 HLA-A*24:02[+] individuals who received the vaccine. For each study participant, antibodies against the spike protein but not against the nucleocapsid protein were observed in blood samples collected after the second vaccine dose (Supplementary Table 1). We stimulated peripheral blood mononuclear cells (PBMCs) from these individuals with 16 SARS-CoV-2 spike peptides that were reported to be HLA-A*24:02-restricted epitopes (Supplementary Table 2) and then cultured them for 14 days. The cultured cells were analyzed using ICS assays to identify CD8[+] T cells that were specific for these epitopes. We found CD8[+] T cells that were specific for 7 of the 16 HLA-A*24:02-restricted epitopes in these 17 participants (Fig. 1a, b). CD8[+] T cells specific for S-QI9, S-NF9, and S-VYF10 were found in PBMCs from 13, 10, and 4 individuals, respectively, while those specific for S-VF12, S-VFF10, S-GL10, and S-RF10 were detected in PBMCs from only one individual (Fig. 1a). These results indicated that S-QI9 and S-NF9 were the immunodominant epitopes in individuals who received the vaccine. T cells specific for two or more HLA-A*24:02-restricted epitopes were found in 11 of 17 individuals (Fig. 1c), while T cells specific for one or two immunodominant epitopes were detected in 15 individuals (Fig. 1d). These findings indicate that the vaccine effectively induced HLA-A*24:02-restricted T cells specific for immunodominant epitopes. We next analyzed correlations between the frequency of S-NF9-specific or S-QI9-specific T cells and the interval since the second vaccine dose. S-NF9-specific T cells were detected only in individuals who were tested within 11 weeks after the second vaccine dose, while S-QI9-specific T cells were found at 21 weeks after the second vaccination (Fig. 1e and Supplementary Fig. 1b). These results imply that the vaccine produced and maintained memory T cells that were specific for these immunodominant epitopes for more than 10 weeks.

We performed a longitudinal analysis of HLA-A*24:02-restricted T cells specific for 4 epitopes in 3 individuals during and after the vaccination protocol. Peripheral blood was collected before the first vaccination, at 2 or 3 weeks after the first vaccination, and at 2 and 4 weeks after the second vaccine dose (Fig. 1f and Supplementary Fig. 1c). S-NF9-specific and S-QI9-specific T cells were found after the second vaccine dose in 3 and 2 individuals, respectively. S-NF9-specific and S-QI9-specific T cells were detected in 1 and 2 individuals, respectively, after the first vaccine dose. However, S-VYF10-specific and S-VF12-specific T cells were observed at 2 weeks after the second

vaccine dose in only one individual (Fig. 1f). These results indicate that T cells specific for immunodominant epitopes, S-NF9- and S-QI9, were elicited even after the first vaccine dose.

Next, we analyzed the frequency of T cells specific for two immunodominant epitopes in PBMCs and compared the frequencies before and after in vitro expansion of these T cells. To detect S-NF9- and S-QI9-specific T cells in PBMCs from vaccinated individuals, we used HLA-A*24:02-tetramers with S-NF9 or S-QI9 peptide. We analyzed PBMCs collected after the second vaccine dose from 15 of 17 individuals (PBMC samples collected from two individuals were not available for this analysis). S-NF9- and S-QI9-specific T cells in PBMCs were clearly detected by the tetramers (Fig. 2a). S-NF9- and S-QI9-specific T cells were detected in 11 and 13 individuals, respectively (Fig. 2b). A strong correlation between the frequencies of tetramer-binding CD8[+] T cells and IFN-γ-producing cells was found in an ICS assay of cultured T cells (Fig. 2c), indicating that these immunodominant-specific T cells were efficiently induced and had the ability to expand in vitro. We analyzed the correlation between the frequency of tetramer-binding CD8[+] T cells in ex vivo PBMCs and the interval since the second vaccine dose (Fig. 2d) and the correlation between the frequency of these same T cells 7 weeks after the first vaccination (4 weeks after the second dose) (Fig. 2e and Supplementary Fig. 2a). These results supported the findings in the analysis of the cultured T cells by ICS assay which were shown in Fig. 1e, f.

### Characterization of HLA-A*24:02-restricted immunodominant T cell epitopes

We investigated the ability of CD8[+] T cells to recognize the seven HLA-A*24:02-restricted SARS-CoV-2 epitopes (sensitivity of CD8[+] T cells for the epitope) by analyzing T cell responses to titrated peptides using ICS assays. S-QI9-specific and S-NF9-specific T cells both showed a maximum response at a specific peptide concentration of 1–10 nM, whereas T cells specific for the other epitopes exhibited maximum responses at peptide levels >1000 nM (Fig. 3 and Supplementary Fig. 1d). Peptide concentrations resulting in approximately 50% of the maximum response were as follows: 0.1 and 1 nM for S-QI9 and S-NF9, respectively, and 10–1000 nM for the other five epitopes. These results indicate that S-QI9-specific and S-NF9-specific T cells have approximately 10–10,000 times higher antigen sensitivity to the epitopes than the other epitope-specific T cell populations.

To determine the effect of epitope peptide binding ability to HLA-A*24:02 on HLA-A*24:02-mediated T cell recognition, HLA stabilization assays were performed using RMA-S cells that expressed HLA-A*24:02. Out of 16 peptides tested, 13 peptides bound to HLA-A*24:02, whereas the peptides S-RF11, S-RF9, and S-YW9 did not bind (Fig. 4a and Supplementary Fig. 3b). A trend toward a positive correlation was observed between the frequency of responders to these 7 epitopes and HLA-A*24:02 binding ability of the peptides except the case of S-RF10 (Fig. 4b). We next investigated affinities of the two immunodominant (S-QI9 and S-NF9) and two subdominant (S-VFF10 and S-VYF10) epitopes for HLA-A*24:02 by using titrated peptides (Fig. 4c and Supplementary Fig. 3c). S-QI9 had an affinity that was approximately 1000 times higher than that of the other three peptides, while S-NF9 had a binding affinity similar to that of the S-VFF10 and S-VYF10 peptides. Together with the data shown in Fig. 3, these results suggested the following mechanisms for high antigen sensitivity displayed by the 2 immunodominant epitope-specific T cell populations. S-NF9-specific T cells have a high affinity T cell receptor (TCR), while S-QI9-specific T cells have a medium affinity TCR but higher sensitivity for the epitope because S-QI9 has a high binding affinity for HLA-A*24:02.

### HLA-A*24:02-restricted CD8[+] T cells elicited in convalescent individuals previously infected with SARS-CoV-2

We analyzed the induction of HLA-A*24:02-restricted spike epitope-specific CD8[+] T cells in HLA-A*24:02[+] individuals previously infected

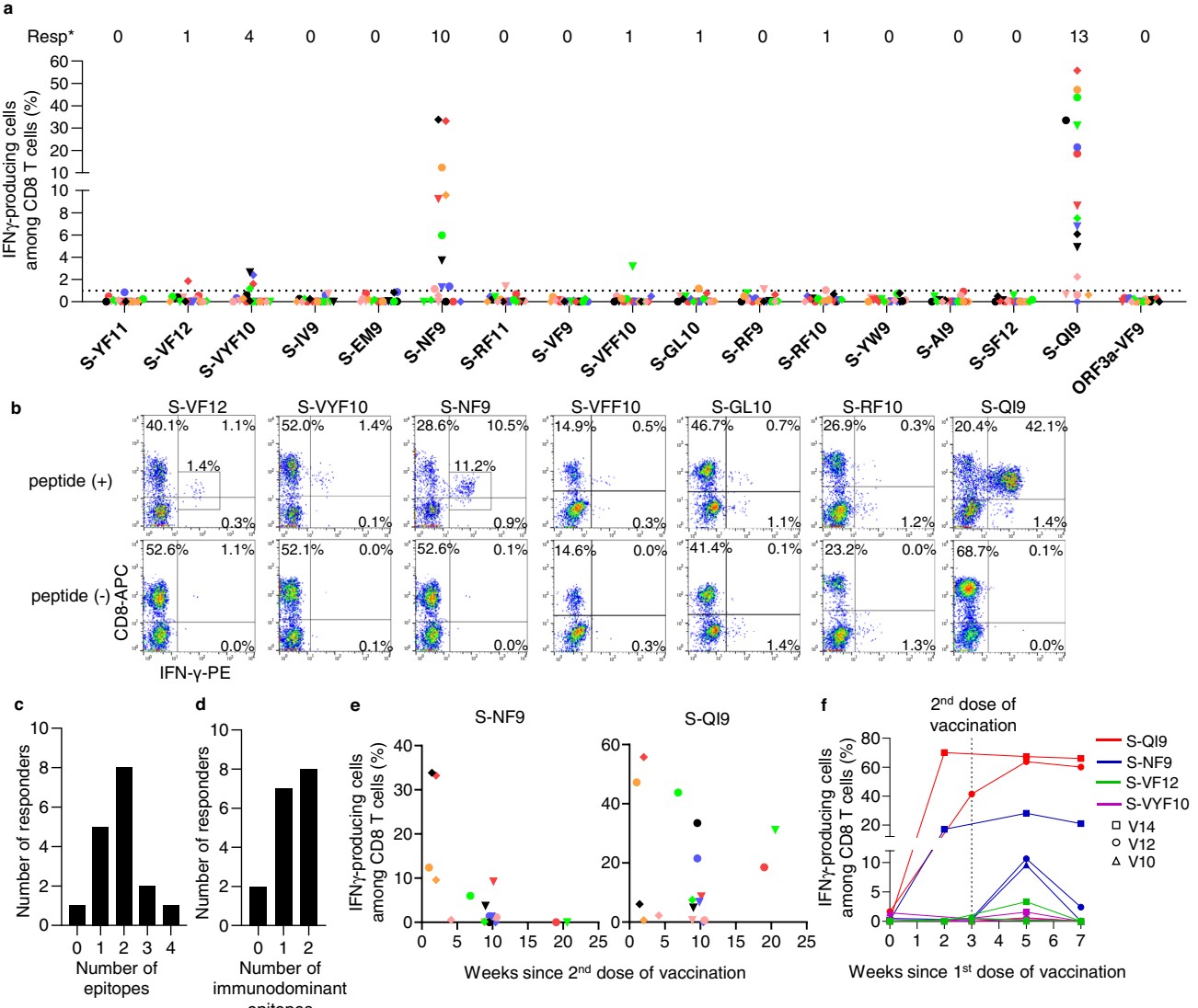

**Fig. 1 | Induction of HLA-A*24:02-restricted CD8+ T cells specific for SARS-CoV-2 spike epitopes in HLA-A*24:02+ individuals vaccinated with an mRNA vaccine.** HLA-A*24:02-restricted T cell responses to SARS-CoV-2 spike epitopes were analyzed in HLA-A*24:02+ individuals who received 2 doses of the BNT162b2 mRNA vaccine. HLA-A*24:02-restricted, SARS-CoV-2 spike-specific bulk T cells were generated by stimulating peripheral blood mononuclear cells (PBMCs) from 17 HLA-A*24:02+, vaccinated individuals with peptide cocktails containing 5 or 6 HLA-A*24:02-restricted SARS-CoV-2 spike peptides or an ORF3a epitope and then culturing the cells for 2 weeks. Responses of the bulk T cells to C1R-A2402 cells pre-pulsed with 1 μM of each peptide were analyzed by ICS assays. **a** Frequency of epitope-specific CD8+ T cells expressing IFN-γ in 17 vaccinated, HLA-A*24:02+ individuals. The dotted line at 1.0% for IFN-γ-producing cells within the CD8+ T cell population represents the threshold for a positive response. Resp: number of responders. **b** Representative results of CD8+ T cell responses to 7 epitope peptides. **c** Number of responders to 0–4 epitopes. **d** Number of responders to 0–2 immunodominant epitopes. **e** Correlation between T cell response to 2 immunodominant epitopes (S-NF9 and S-QI9) and the number of weeks since the second vaccination in 17 HLA-A*24:02+ individuals. The 17 individuals were discriminated by different symbols and colors. **f** Longitudinal analysis of T cell response to 2 immunodominant and 2 subdominant epitope peptides after the first vaccine dose. Responses of the bulk T cells established from 3 HLA-A*24:02+ vaccinated individuals to C1R-A2402 cells pre-pulsed with 1 μM of S-QI9, S-NF9, S-VF12, and S-VYF10 peptides were analyzed at 4 time points (0, 2–3, 5, and 7 weeks) after the first vaccine dose using ICS assays. Gating strategy in flowcytometry analysis is shown in Supplementary Fig. 1a. Source data are provided as a Source Data file.

with SARS-CoV-2. PBMCs from 10 HLA-A*24:02+ individuals were collected 2–50 weeks after diagnosis of SARS-CoV-2 infection (Supplementary Table 3). These PBMCs were analyzed by both an ICS assay of cells cultured for 2 weeks and a tetramer binding assay of ex vivo PBMCs. The ICS assay of cultured cells detected S-NF9-specific and S-QI9-specific T cells in four and three individuals, respectively, whereas T cells specific for the other seven spike epitopes were detected at low frequency (1–2% among CD8+ T cells) at a small number of the individuals (Fig. 5a and Supplementary Fig. 4a). These results indicated that S-NF9-specific and S-QI9-specific T cells are also immunodominant T cells in convalescent individuals. The tetramer-binding assay demonstrated that S-NF9-specific and S-QI9-specific T cells were also

detected in ex vivo PBMCs from 4/10 and 6/10 individuals, respectively (Fig. 5b and Supplementary Fig. 2b). Compared with vaccinated individuals, convalescent individuals generally showed a lower frequency of these immunodominant epitope-specific T cells. The tetramer-binding assay also detected these immunodominant T cells in convalescent individuals 2–50 weeks after diagnosis of SARS-CoV-2 infection (Fig. 5c and Supplementary Fig. 2b). T cells derived from individuals within 30 weeks of diagnosis of COVID-19 infection could expand in vitro, whereas those derived from individuals after 30 weeks of diagnosis could not (Fig. 5d and Supplementary Fig. 4a). This finding implies that these immunodominant memory T cells derived from convalescent individuals within 30 weeks of diagnosis maintain their

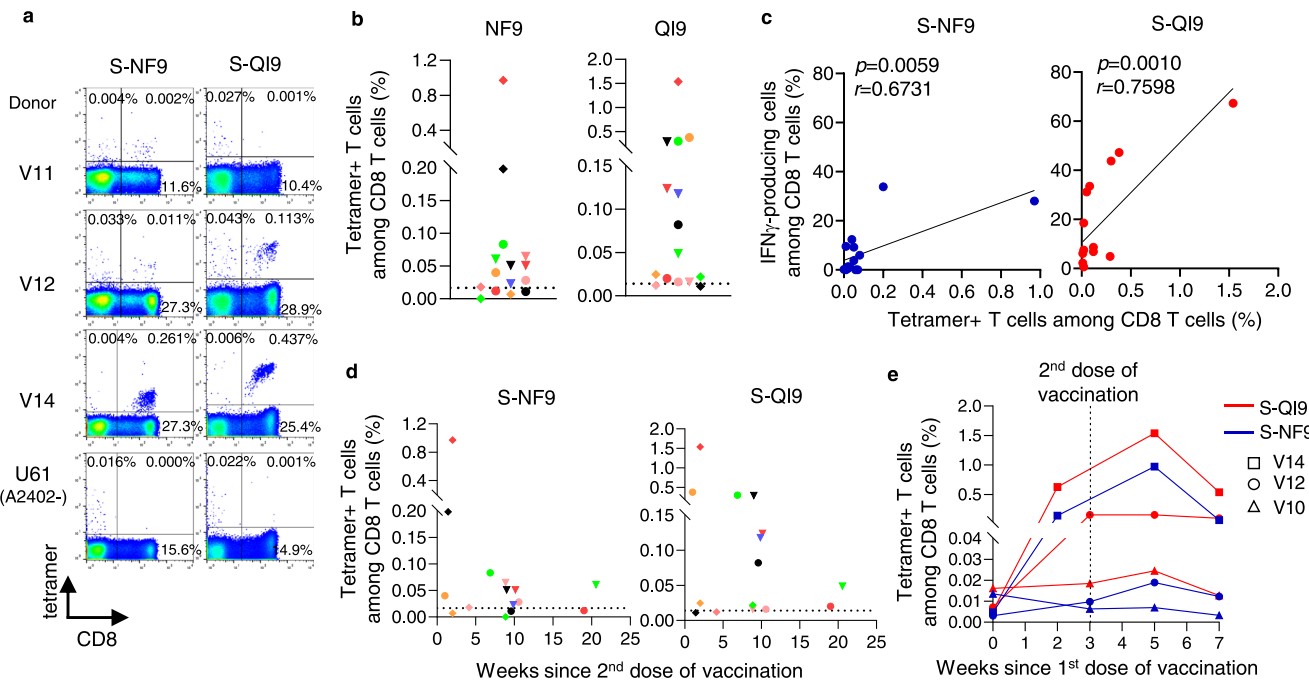

**Fig. 2 | Analysis of CD8⁺ T cells specific for two immunodominant epitopes in ex vivo PBMCs from HLA-A*24:02⁺ individuals vaccinated with an mRNA vaccine. a** Representative results of HLA-A*24:02-S-NF9-tetramer⁺ or HLA-A*24:02-S-QI9-tetramer⁺ CD8⁺ T cells in three vaccinated HLA-A*24:02⁺ individuals. U61 is a HLA-A*24:02-negative control. **b** Frequency of HLA-A*24:02-S-NF9-tetramer⁺ or HLA-A*24:02-S-QI9-tetramer⁺CD8⁺ T cells in ex vivo PBMCs from 15 HLA-A*24:02⁺ individuals corresponding to samples used in Fig. 1a. The dotted line represents the threshold for positive binding. **c** Correlation between the frequency of tetramer-binding CD8⁺ T cells and that of IFN-γ-producing cells in an ICS assay. Statistical analyses were performed using two-tailed Pearson's correlation test. **d** Correlation between tetramer-binding CD8⁺ T cells and the number of weeks since the second vaccination in 15 HLA-A*24:02⁺ individuals. **e** Frequency of HLA-A*24:02-NF9-tetramer⁺ or HLA-A*24:02-S-QI9-tetramer⁺ CD8⁺ T cells. The samples used in Fig. 1f were used for this analysis. Gating strategy in flowcytometry analysis is shown in Supplementary Fig. 2a. Source data are provided as a Source Data file.

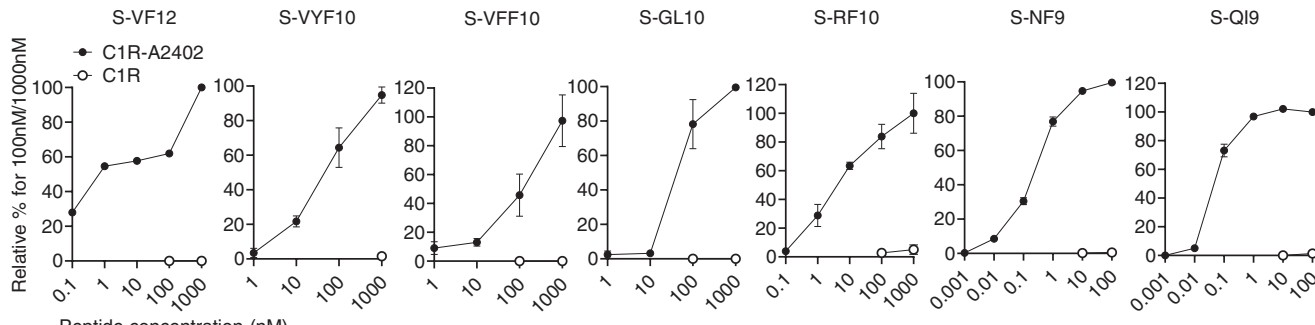

**Fig. 3 | Ability of HLA-A*24:02-restricted CD8⁺ T cells specific for spike epitopes to recognize epitope peptides.** Recognition of 7 HLA-A*24:02-restricted spike epitope peptides by SARS-CoV-2-specific CD8⁺ T cells from HLA-A*24:02⁺, vaccinated individuals. T cell responses of bulk cultured T cells were analyzed using ICS assays after exposure to C1R-A*2402 or C1R cells pre-pulsed with various concentrations of peptides. Relative % response to maximum response at 100 or 1000 nM were presented. The data are presented as the mean and SD (*n* = 3) except for S-VF12-specific T cells. Gating strategy in flowcytometry analysis is shown in Supplementary Fig. 1a. Source data are provided as a Source Data file.

capability to expand in vitro after stimulation with the epitope, whereas this capability was lost after 30 weeks from diagnosis.

## Vaccine-induced long-term memory CD8⁺ T cells specific for HLA-A*24:02-restricted immunodominant epitopes

S-QI9-specific T cells were detected in 2 individuals at approximately 20 weeks after the second vaccine dose (Figs. 1e and 2d), suggesting that the vaccine elicited a long-term memory CD8⁺ T cell response specific for S-QI9. To clarify the existence of long-term memory CD8⁺ T cells in vaccinated HLA-A*24:02⁺ individuals, we collected PBMCs from 14 individuals who had T cells specific for at least one of the three epitopes, S-QI9, S-NF9, and S-VYF10, at the first point of

analysis shown in Fig. 1a, and then investigated CD8⁺ T cells specific for these epitopes. The T cells from these individuals were analyzed during 11–22 weeks after the first point of analysis by ICS assay of cultured T cells (Fig. 6a and Supplementary Fig. 4b). S-VYF10-specific T cells were detected at both time points in two individuals, while they were found at only the first time point in one individual. S-NF9-specific T cells were found at both time points in 8 individuals, at only the first time point in 1 individual, and at only the second time point in 2 individuals. S-QI9-specific T cells were found at both time points in 13 individuals. The frequency of S-QI9-specific or S-NF9-specific T cells was similar at both time points in most individuals who had these T cells. These results indicated that the mRNA vaccine elicited

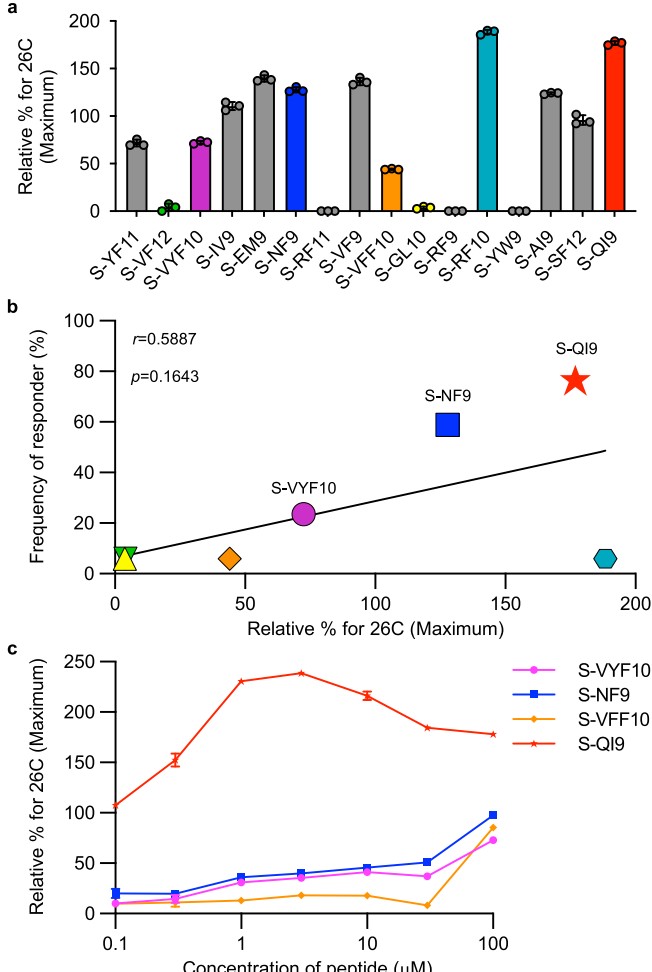

**Fig. 4 | Binding of HLA-A*24:02-restricted spike epitope peptides to HLA-A*24:02. a** Binding of 16 HLA-A*24:02-restricted SARS-CoV-2 epitope peptides to HLA-A*24:02 was measured by HLA class I stabilization assays using RMA-S-A2402 cells and a peptide concentration of 100 μM. The colored bar indicates T cell epitopes in vaccinated, HLA-A*24:02⁺ individuals. Relative % for 26 °C (Maximum) was calculated as shown in Methods. The data are presented as the mean and SD (*n* = 3). **b** Correlation between the binding of 7 epitope peptides to HLA-A*24:02 and frequency of responders to these peptides. Statistical analyses were performed using two-tailed Pearson's correlation test. **c** Binding of 4 epitope peptides to HLA-A*24:02 was analyzed by HLA stabilization assays using peptide. concentrations ranging from 0.1 to 100 μM. The binding results in **a, c** are plotted as the % increase in surface HLA-I expression relative to expression at 26 °C and calculated as the ratio of the mean fluorescence intensity (MFI) of peptide-pulsed RMA-S-A2402 cells compared with that of the control (non-peptide-pulsed) cells at 26 °C. The data are presented as the mean and SD (*n* = 3). Gating strategy in flowcytometry analysis is shown in Supplementary Fig. 3a. Source data are provided as a Source Data file.

long-term memory CD8⁺ T cells specific for S-NF9, S-QI9, and S-VYF10 epitopes and maintained them for >30 weeks after the second vaccine dose in 8 individuals (Fig. 6a). The frequency of S-QI9-specific and S-NF9-specific T cells in ex vivo PBMCs at the second time point was also evaluated by staining PBMCs with the HLA-A*24:02-tetramers. S-QI9-specific and S-NF9-specific T cells were detected in 12/14 and 8/14 individuals tested, respectively (Fig. 6b and Supplementary Fig. 2c). We found a strong correlation between the frequencies of tetramer-binding CD8⁺ T cells and IFN-γ-producing cells in an ICS assay (Fig. 6c). These findings indicate that T cells specific for the two immunodominant epitopes were maintained for >30 weeks after the second vaccine dose in eight individuals and retained the capability to expand in vitro.

## Cross-recognition of S-QI9 variant peptides derived from seasonal human coronaviruses by S-QI9-specific T cells

We analyzed the ability of S-QI9-specific T cells to cross-recognize S-QI9 mutant peptides derived from 4 seasonal human coronaviruses (NL63, 229E, OC43, and HKU1). Three amino acid substitutions were found in the S-QI9 peptide from these seasonal viruses compared with S-QI9 from SARS-CoV-2 (Fig. 7a). We investigated the capability of S-QI9-specific bulk T cells from 13 HLA-A*24:02⁺ individuals to cross-recognize C1R-A2402 cells that were pre-pulsed with 1 nM of the variant or QI9 peptides by performing ICS assays. All 13 S-QI9-specific T cell populations effectively recognized HKU1 and OC43 variant peptides, whereas these T cells demonstrated different recognition patterns for NL63 and 229E variant peptides (Fig. 7a and Supplementary Fig. 4c). Bulk T cells from 4 individuals exhibited no or very weak recognition of NL63 and 229E variant peptides, and T cells from 2 individuals uniformly recognized these variant peptides and SARS-CoV-2 S-QI9. The T cells from the other 7 individuals moderately recognized NL63- and 229E-derived variant peptides. Next, we analyzed cross-recognition by T cells from 4 individuals that had shown 3 different patterns of recognition after the 1 nM peptide pre-pulse (Fig. 7b and Supplementary Fig. 4d). S-QI9-specific T cells from all 4 individuals recognized OC43 and HKU1 peptides to a greater extent than the SARS-CoV-2 S-QI9 peptide and showed different recognition patterns for NL63 and 229E peptides. The T cells from study participant V14 recognized neither the NL63 nor 229E peptide. The T cells from individuals V2, V6, and V8 similarly recognized the 229E peptide and showed weaker recognition of NL63. Overall, these results indicated that S-QI9-specific T cells from the vaccinated individuals had higher sensitivity for OC43 and HKU1 peptides than the SARS-CoV-2 S-QI9 epitope but much lower sensitivity for NL63 and 229E peptides.

We investigated the affinity of these peptides for HLA-A*24:02 by performing HLA stabilization assays (Fig. 7c and Supplementary Fig. 3d) and measured the stability of HLA-A*24:02-peptide complexes by performing HLA-peptide complex dissociation assays (Fig. 7d and Supplementary Fig. 3e). OC43 and HKU1 peptides had approximately 10–30 times weaker affinity for HLA-A*24:02 than that of S-QI9. The stability of the HLA-A*24:02-OC43 peptide complex was identical to that of the HLA-A*24:02-S-QI9 peptide complex. The stability of the HLA-A*24:02-HKU1 peptide complex was weaker than that of the HLA-A*24:02-S-QI9 peptide complex. These findings together suggested that HLA-A*24:02 can present the S-QI9 peptide more effectively than either the OC43 or HKU1 peptides. However, the 229E peptide had an affinity similar to that of the OC43 and HKU1 peptides, while the stability of the HLA-A*24:02-229E peptide complex was weaker than that of the HLA-A*24:02-HKU1 or HLA-A*24:02-OC43 peptide complexes. The NL63 peptide had a substantially weaker affinity and stability for HLA-A*24:02 than the other 3 peptides. Together, these findings suggested that HLA-A*24:02 can present the 229E and NL63 peptides much less effectively than the OC43 and HKU1 peptides. Furthermore, S-QI9-specific T cells possessed a TCR with a higher affinity for OC43 and HKU1 peptides than S-QI9. A lower antigen sensitivity of S-QI9-specific T cells for 229E and NL63 peptides may result from lower HLA-A*24:02 affinity and complex stability of these peptides than that observed for S-QI9.

## Discussion

Because HLA-A*24:02 is a common HLA allele worldwide, the analysis of HLA-A*24:02-restricted SARS-CoV-2-specific T cells is important to understand CD8⁺ T cell immunity during and after SARS-CoV-2 infection. Indeed, many HLA-A*24:02-restricted SARS-CoV-2 epitopes have been reported or proposed (Supplementary Table 2), although the majority of T cell epitopes were identified in individuals infected with SARS-CoV-2 by performing the ex vivo activation-induced cell marker (AIM), ELISpot, or multimer binding assay (Supplementary Table 2). In

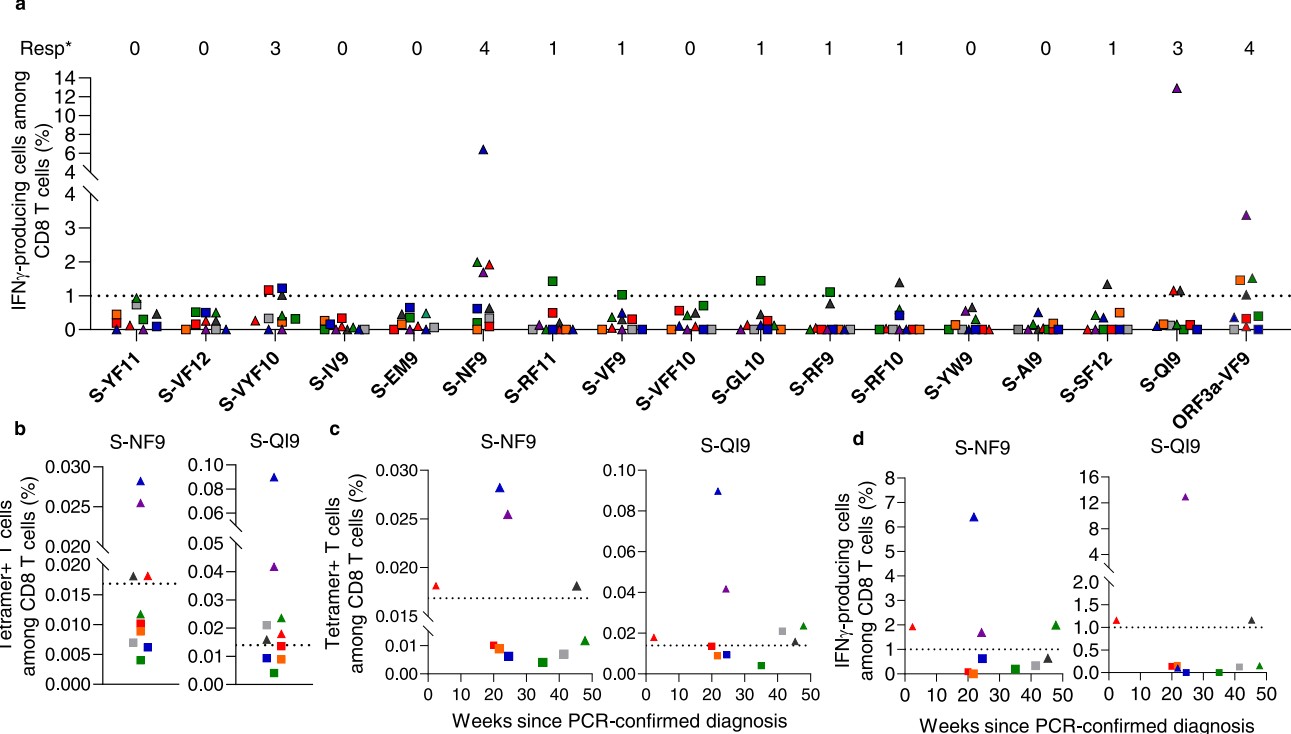

**Fig. 5 | HLA-A*24:02-restricted CD8⁺ T cells elicited in individuals previously infected with SARS-CoV-2. a** Frequency of epitope-specific CD8⁺ T cells expressing IFN-γ in 10 HLA-A*24:02⁺ individuals previously infected with SARS-CoV-2. The dotted line at 1.0% for IFN-γ-producing cells within the CD8⁺ T cell population represents the threshold for a positive response. Resp: number of responders. **b** Frequency of HLA-A*24:02-S-NF9-tetramer⁺ or HLA-A*24:02-S-QI9-tetramer⁺ CD8⁺ T cells in ex vivo PBMCs from 10 HLA-A*24:02⁺ individuals. The dotted line represents the threshold for positive binding. **c** Correlation between the tetramer-binding CD8⁺ T cells and the number of weeks since the diagnosis of infection by PCR in 10 HLA-A*24:02⁺ individuals. **d** Correlation between the T cell response to two immunodominant epitopes (S-NF9 and S-QI9) and the number of weeks since the diagnosis of infection by PCR in 10 HLA-A*24:02⁺ individuals. The 10 individuals are discriminated by different symbols and colors, and these symbols and colors are consistent between figure panels (**b**–**d**). Gating strategy in flowcytometry analysis is shown in Supplementary Fig. 1a (ICS assay) and Supplementary Fig. 2a (tetramer binding). Source data are provided as a Source Data file.

the present study, we attempted to detect HLA-A*24:02-restricted CD8⁺ T cells that recognized 16 reported spike protein epitopes by culturing T cells with the epitope peptides and performing ICS assays using CIR-A2402 cells as stimulator cells. We identified 7 specific epitopes that induced HLA-A*24:02-restricted CD8⁺ T cells in vaccinated individuals. The analysis using the HLA stabilization assay demonstrated that 3 peptides did not bind to HLA-A*24:02. These 3 non-binding peptides were previously reported as HLA-A*24:02 epitopes in a study using an AIM assay[49]. These results indicate that T cell responses to these 3 reported HLA-A*24:02-restricted SARS-CoV-2 spike epitopes were restricted by other HLA class I alleles rather than HLA-A*24:02. Thus, the present study indicates that some reported SARS-CoV-2 epitopes are not optimal epitopes or that their HLA-restriction phenotype is incorrect.

Recent studies showed that T cells specific for these epitopes were frequently detected among individuals infected with SARS-CoV-2[49–55]. The present study demonstrated that S-QI9 and S-NF9 were recognized as the immunodominant epitopes in the vaccinated and the convalescent individuals. Thus, HLA-A*24:02-restricted T cells specific for these immunodominant epitopes were effectively elicited in both vaccinated and SARS-CoV-2-infected individuals. The findings of the current study suggested that these immunodominant memory T cells derived from convalescent individuals more than 30 weeks after diagnosis might lose the capability to expand in vitro after stimulation with the epitope, implying that the long-term memory T cells elicited after SARS-CoV-2 infection might maintain the capability to proliferate in convalescent individuals only up to 30 weeks after infection. Further analysis of a large number of convalescent individuals will clarify whether long-term memory T cells elicited after infection can generally

maintain this capacity to proliferate for longer periods of time (e.g., 6 months).

The S-QI9- and S-NF9-specific T cells had higher sensitivity for epitope peptides compared with other HLA-A*24:02-restricted T cells. Interestingly, both T cell populations had a similar level of sensitivity for their specific epitopes, while the S-QI9 peptide had much stronger affinity for HLA-A*24:02 than S-NF9. These findings indicated that S-NF9-specific T cells had a higher affinity TCR than S-QI9-specific cells, while the S-QI9-specific T cells had higher sensitivity for its epitope since S-QI9 is very high affinity peptide for HLA-A*24:02. T cells possessing high sensitivity for epitope are expected to have an increased ability to recognize SARS-CoV-2-infected cells and to suppress replication of SARS-CoV-2. S-QI9 is a conserved epitope among circulating SARS-CoV-2 variants, including Omicron (https://www.gisaid.org)[56,57], while the Delta and the Omicron BA.5 variants have one mutation at the fifth position of S-NF9 (Supplementary Table 4). A recent study demonstrated that S-NF9-specific T cells failed to recognize the variant peptide derived from the Delta variant[58]. Together, these findings suggest that S-QI9-specific T cells have strong ability to eradicate cells infected with all variant viruses, including the Omicron BA.5 and the Delta ones, while S-NF9-specific T cells may be effective against most variants except for the Delta and the Omicron BA.5 variants.

A recent study showed that S-QI9-specific T cells were induced in PBMCs from HLA-A*24:02⁺ individuals who had not been exposed to SARS-CoV-2 when these cells were stimulated with the S-QI9 peptide and further demonstrated that these S-QI9-specific T cells cross-recognized S-QI9 variants derived from the 4 seasonal human coronaviruses NL63, 229E, OC43, and HKU1 with different recognition patterns[59]. The patterns of cross-recognition included (1) equal cross-

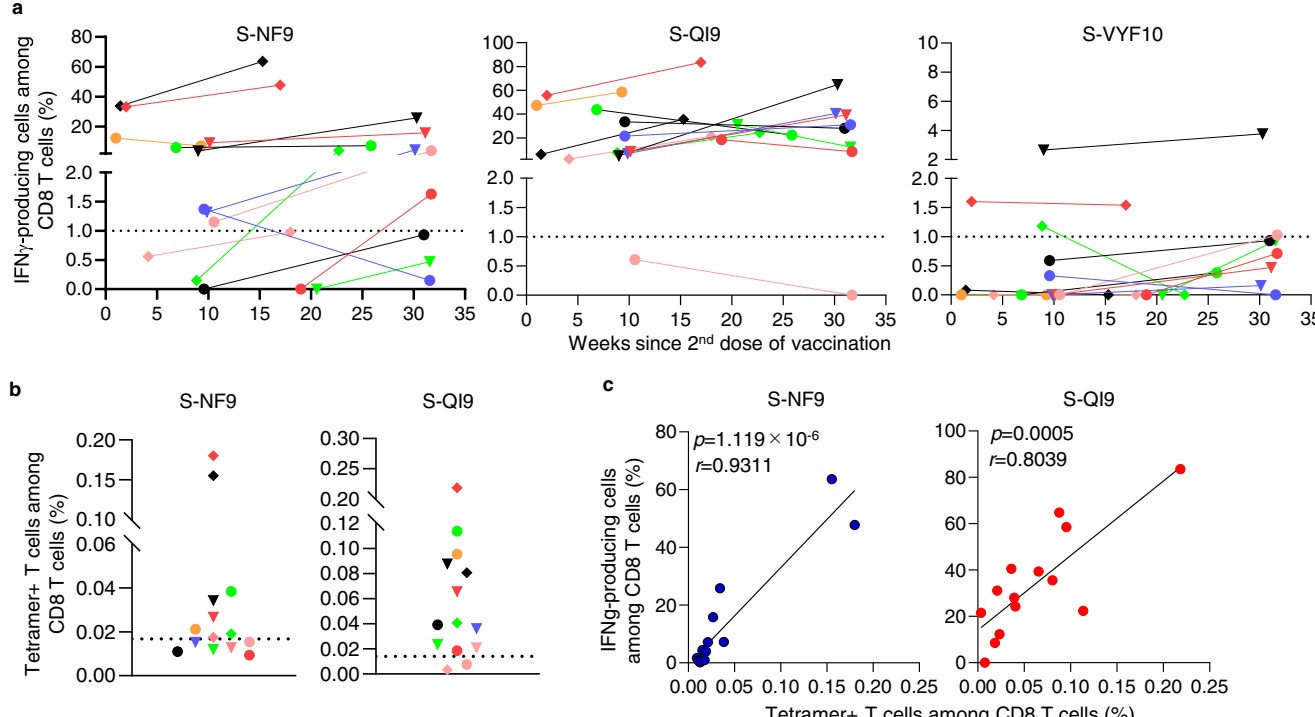

**Fig. 6 | Induction and maintenance of long-term memory CD8⁺ T cells specific for immunodominant epitopes. a** Longitudinal analysis of HLA-A*24:02-restricted T cell responses to 2 immunodominant and 1 subdominant spike epitopes of SARS-CoV-2 after the second vaccine dose. The responses of SARS-CoV-2-specific bulk T cells established from 14 HLA-A*24:02⁺, vaccinated individuals were analyzed at 2 time points using ICS assays after exposure to C1R-A2402 cells that were pre-pulsed with 1 µM of NF9, QI9, and VYF10 peptides. The dotted line at 1.0% for IFN-γ-producing cells among CD8⁺ T cell population represents the threshold for a positive response. **b** Frequency of HLA-A*24:02-S-NF9-tetramer⁺ or HLA-A*24:02-S-QI9-tetramer⁺ CD8⁺ T cells in ex vivo PBMCs from 14 HLA-A*24:02⁺ individuals corresponding to the samples used in figure panel a. The dotted line represents the threshold for positive binding. **c** Correlation between the frequency of tetramer-binding CD8⁺ T cells and that of IFN-γ-producing cells. Statistical analyses were performed using two-tailed Pearson's correlation test. Gating strategy in flowcytometry analysis is shown in Supplementary Fig. 1a (ICS assay) and Supplementary Fig. 2a (tetramer binding). Source data are provided as a Source Data file.

recognition for S-QI9 variant epitopes from 4 seasonal human coronaviruses, (2) no cross-recognition for all 4 S-QI9 variants, (3) higher cross-recognition for NL63 and 229E variants and lower cross-recognition for OC43 and HKU1, and (4) lower cross-recognition for NL63 and 229E variants and higher cross-recognition for OC43 and HKU1. However, in the present study, we revealed that S-QI9-specific T cells derived from all of the vaccinated HLA-A*24:02⁺ individuals cross-recognized the OC43 and HKU1 variants to a greater extent than S-QI9 and showed different cross-recognition patterns for the NL63 and 229E variants (no recognition of both variants, equal cross-recognition for the 229E variant but lower cross-recognition for NL63, and lower cross-recognition for both 229E and NL63 variants). These findings indicate that different S-QI9-specific T cell repertoires were elicited in the unexposed and vaccinated individuals and suggest that the vaccine-induced T cell repertoire may be less diverse than the T cell populations induced by seasonal human coronaviruses. From these findings, it is speculated that the vaccine might induce cross-reacting T cells from T cell population comprising the pre-existing T cells induced by seasonal human coronaviruses.

A previous study detected pre-existing memory T cells with protective ability against SARS-CoV-2 in SARS-CoV-2-unexposed healthy individuals and speculated that these T cells would expand in vivo and contribute to viral control, preventing infection and decreasing mortality[60]. These pre-existing T cells may contribute to protection against infection in vaccinated individuals if the vaccine can induce expansion of these T-cells. The current study, together with a previous study[59], showed that T cells that cross-recognize the S-QI9 epitope in SARS-CoV-2 and seasonal human coronaviruses pre-exist in SARS-CoV-2-unexposed healthy individuals. It is therefore interesting to analyze the correlation between pre-existing T cells that cross-recognize the S-QI9 epitope and the expansion of these T cells after vaccination. In the current study, we only analyzed the pre-existing T cells in three individuals because of sample limitations, and found that two individuals had pre-existing T cells that cross-recognized the S-QI9 epitope. In these two individuals, S-QI9-specific T cells were strongly and weakly induced after vaccination, respectively (Figs. 1f and 2e). To fully interpret the role of pre-existing T cells, further analyses are needed to clarify the correlation between the presence of pre-existing T cells and the induction effect of these T cells after vaccination, as well as the role of the pre-existing T cells in aborting infection and decreasing mortality.

The S-QI9 peptide displayed a higher affinity for and stability with HLA-A*24:02 than that observed for the variant peptides derived from the 4 seasonal coronaviruses. Substitutions at P1 and P9 distinguish S-QI9 from the variant peptides and may affect peptide binding affinity for and stability with HLA-A*24:02. Recent crystallographic analysis of the HLA-A*24:02-S-QI9 peptide complex demonstrated that P2, P3, P6, and P9 were involved in the HLA-A*24:02-peptide interaction, while the side chains of amino acids at P1, P4, P5, P7, and P8 were directed toward the solvent region, suggesting that these side chains may directly interact with the TCR[59]. Therefore, the substitution from Val to Ile at P9 may determine the differences in HLA-A*24:02 binding affinity and stability between S-QI9 and peptides from seasonal coronaviruses while the substitutions at P1 and P8 between SARS-CoV-2/HKU-1/OC43 and NL63/229E may increase TCR recognition of peptides derived from HKU-1 and OC43.

In the current study, we demonstrated that long-term memory CD8⁺ T cells specific for two SARS-CoV-2 spike epitopes were

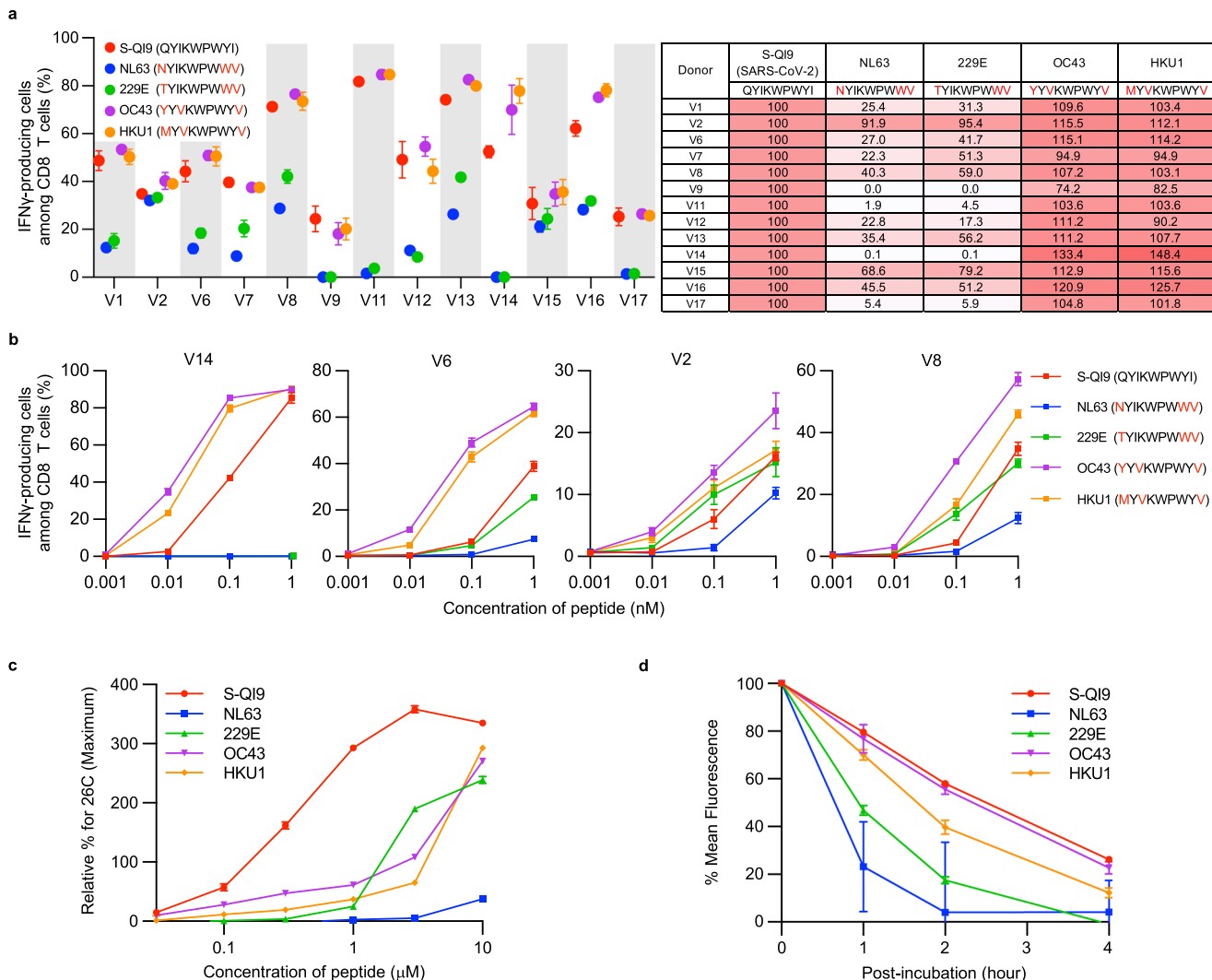

**Fig. 7 | Recognition of S-QI9 variants peptides derived from seasonal human coronaviruses by S-QI9-specific T cells and binding affinities and stabilities of S-QI9 variant peptides for HLA-A*24:02. a** Left: Recognition of S-QI9 and S-QI9 mutant peptides derived from 4 seasonal human coronaviruses (NL63, 229E, OC43, and HKU1) by S-QI9-specific CD8+ T cells established from 13 HLA-A*24:02+ individuals who were vaccinated with 2 doses of the mRNA vaccine. Responses of bulk T cells to C1R-A*2402 cells pre-pulsed with 1 nM of each peptide were analyzed by ICS assays. Right: The frequencies of CD8+ T cells specific for S-QI9 variant peptides relative to CD8+ T cells specific for S-QI9 peptides. **b** Recognition of S-QI9 and 4 S-QI9 variant peptides derived from seasonal human coronaviruses by S-QI9-specific CD8+ T cells established from 4 representative HLA-A*24:02+ vaccinated individuals. T cell responses were analyzed by ICS assays after exposure to C1R-A*2402 cells pre-pulsed with various concentrations of peptides. **c** Binding of the S-QI9 and 4 S-QI9 mutant peptides to HLA-A*24:02 was analyzed by HLA stabilization assays using peptide concentrations ranging from 0.03 to 10 μM. Binding results are plotted as the % increase in surface HLA-I expression relative to expression at 26 °C and calculated as the ratio of the mean fluorescence intensity (MFI) of peptide-pulsed RMA-S-A2402 cells compared with that of the control (non-peptide-pulsed) cells at 26 °C. **d** Stabilization of the HLA-A*24:02-S-QI9 peptide or HLA-A*24:02-S-QI9 mutant peptide complexes. The data shown in **a–d** are presented as the mean and SD (*n* = 3). Gating strategy in flowcytometry analysis is shown in Supplementary Fig. 1a (ICS assay) and Supplementary Fig. 3a (peptide binding). Source data are provided as a Source Data file.

maintained for at least 30–31 weeks after the second vaccine dose and retained the ability to expand in vitro. These findings suggested the important role of these HLA-A*24:02-restricted T cells in SARS-CoV-2 vaccination. Although the HLA-A*24:02 is one of the most common HLA alleles worldwide, this allele is more frequently found in Asia. Approximately 65% of the population have this allele in Japan[61–63]. Therefore, the effects of S-QI9-specific or S-NF9-specific T cells in SARS-CoV-2 infection may be more pronounced in Asian countries. S-QI9 variants derived from NL63/229E and HKU-1/OC43 seasonal human coronaviruses represent consensus sequences among α/γ and β seasonal coronaviruses, respectively (Fig. 8). Because these S-QI9 variant sequences are conserved among coronaviruses from human and animals, it is expected that SARS-coronaviruses that develop in the future will maintain these variant sequences. If sequences are

maintained, in addition to the preventive effects against SARS-CoV-2 infections, including the Omicron and Delta variants, S-QI9-specific T cells may be key factors in preventing infection and decreasing mortality caused by novel coronaviruses.

## Methods

### Participants

We recruited 17 HLA-A*24:02+ individuals who received two doses of the Pfizer-BioNTech BNT162b2 mRNA vaccine with a three-week interval and 10 HLA-A*24:02+ individuals who had been infected with SARS-CoV-2. The details of the individuals were shown in Supplementary Tables 1 and 3. HLA types of these individuals were determined by standard sequence-based genotyping. This study was approved by the Ethics Committees of Kumamoto University and the National Center

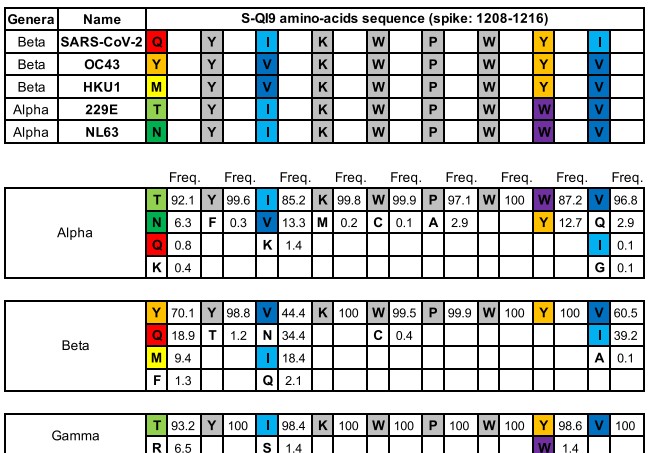

**Fig. 8 | Frequency of amino-acid residues within S-QI9 epitopes from the three genera of coronaviruses.** The upper panel shows the homology of the reported sequence in the S-QI9 epitope region of SARS-CoV-2 compared with that in four seasonal human coronaviruses (OC43, HKU1, 229E, and NL63). Identical residues are color-coded in gray. The lower panel shows the frequency of amino acid variations in alpha-, beta-, and gamma-coronavirus sequences in the S-QI9 epitope region (approximately 2600, 1600, and 630 variations, respectively). These frequencies were calculated using the sequences that were submitted to the NCBI Virus database (https://www.ncbi.nlm.nih.gov/labs/virus/vssi/#/) as spike glycoprotein or spike protein from alpha- (taxid: 693996), beta- (taxid: 694002), and gamma- (taxid: 694013) coronaviruses. These sequences contain variations derived from humans, bats, swine, ferrets, and other animals.

for Global Health and Medicine. Informed consent was obtained from all individuals according to the Declaration of Helsinki. Peripheral blood mononuclear cells (PBMCs) were separated from whole blood.

### Cells

The C1R cell line was maintained in RPMI medium containing 10% fetal calf serum (FCS, R10). C1R cells expressing HLA-A*24:02 (C1R-A2402) and TAP2-deficient RMA-S cells expressing HLA-A*24:02 (RMA-S-A2402) were previously generated by transfecting C1R cells and RMA-S cells, respectively, with the HLA-A*24:02 gene[64,65]. These cells were maintained in R10 containing 0.15 mg/mL hygromycin B.

### Intracellular cytokine staining (ICS) assay

$10^6$ PBMCs from each individual were stimulated with 1µM of a peptide cocktail including 5 or 6 reported HLA-A*24:02-restricted SARS-CoV-2 epitope peptides (Supplementary Table 2) and then cultured in R10 containing IL-2 (20 ng/ml, ProSpec) for 14 days. C1R-A2402 or C1R cells pre-pulsed with epitope peptide were co-cultured with a bulk cultured T cells in a 96-well plate for 4 h at 37 °C in the presence of brefeldin A (10 µg/ml, Sigma-Aldrich). Subsequently, these cells were fixed with 4% paraformaldehyde and incubated in permeabilization buffer [0.1% saponin-10% FBS-phosphate-buffered saline (PBS)] after which they were stained with the reagents of a LIVE/DEAD™ Fixable Near-IR Dead Cell Stain kit (Invitrogen) and APC-conjugated anti-CD8 mAb (BioLegend) followed by PE-conjugated anti-IFN-γ mAb (BioLegend). Detailed information of antibodies used in this study is described in Supplementary Table 5. The percentage of IFN-γ+ cells among the CD8+ T cell population was determined by use of the FACS Canto II (FACS Diva software v9.0). FlowJo v10 software was used for the analyses. A representative flow cytometry gating strategy for IFN-γ+ cells among CD8+ T cells are shown in Supplementary Fig. 1a. Non-specific production of cytokines was excluded by subtracting the data of the negative control, which was the same sample stimulated cells without the peptide and stained with the same mAbs.

### Tetramer staining

Biotinylated HLA-A*24:02 molecules were previously generated[66]. S-QI9 or S-NF9 peptides were added to the refolding solution containing the biotinylation sequence-tagged extracellular domain of the HLA-A*24:02 molecule and β2-microglobulin. The purified monomer complexes were mixed with PE- or APC- labeled streptavidin (Invitrogen). PBMCs were stained with tetramers at 37 °C for 30 min. The cells were then washed twice with R10, followed by staining with FITC-conjugated-anti-CD8 mAb (DAKO), and the reagents of a LIVE/DEAD™ Fixable Near-IR Dead Cell Stain kit (Invitrogen) at 4 °C for 30 min. The cells were washed twice with R10. Detailed information of antibodies used in this study is described in Supplementary Table 5. Data were analyzed with a FACS Canto II instrument (FACS Diva software v9.0). FlowJo v10 software was used for the analyses. A representative flow cytometry gating strategy for tetramer+ cells among CD8+ T cells is shown in Supplementary Fig. 2a. A mean + 3 SD of the frequency of tetramer-binding CD8+ T cells from 5 HLA-A*24:02-negtive individuals for S-QI9 and S-NF9 were 0.0139 and 0.0168%, respectively. Therefore, we defined a positive frequency of S-QI9- and S-NF9-tetramer-binding CD8+ T cells as 0.0139 and 0.0168%, respectively.

### HLA stabilization assay using RMA-S-A2402 cells

These RMA-S transfectant cells were cultured at 26 °C for 16 h and then pulsed with peptides at 26 °C for 1 h and subsequently incubated at 37 °C for 3 h. Staining of cell-surface HLA molecules was performed by using anti-HLA-A11 and HLA-A24 monoclonal antibody (mAb), A11.1 M, and APC-conjugated sheep anti-mouse IgG (Jackson ImmunoResearch). Detailed information of antibodies in this study is described in Supplementary Table 5. Relative % for 26 °C (Maximum) was calculated as follows: [MFI (mean fluorescence intensity) of RMA-S cells pre-pulsed with peptide − MFI of RMA-S cells without peptide pulsing]/[MFI of RMA-S cells incubated at 26 °C − MFI of RMA-S cells without peptide pulsing].

### Peptide−MHC Complex Dissociation Assay using RMA-S-A2402 cells

The cells were cultured at 26 °C for 16 h and then continued to be incubated at 26 °C for 1 h in the presence of 10 µM peptides or in medium only as a negative control. These cells were incubated at 37 °C for 3 h, subsequently washed three times with cold culture medium, and thereafter suspended in culture medium. After 0, 1, 2, and 4 h, samples were taken, washed twice, and immediately stained with HLA-A24 monoclonal antibody (mAb), A11.1 M, for 30 min, washed, and stained with APC-conjugated sheep anti-mouse IgG (Jackson ImmunoResearch). Detailed information of antibodies in this study is described in Supplementary Table 5. The decay of HLA-A*24:02-peptide complexes was determined as the percent mean fluorescence intensity (MFI) remaining: $(MFI_{t(+pep)} − MFI_{t(−pep)})/(MFI_{t=0(+pep)} − MFI_{t=0(−pep)})$. All stained RMA-S-A2402 cells were analyzed with a FACS Canto II instrument (FACS Diva software v9.0). FlowJo v10 software was used for the analyses. A representative flow cytometry gating strategy is shown in Supplementary Fig. 3a.

### Statistics

Statistical analyses of correlation between binding of 7 epitope peptides to HLA-A*24:02 and frequency of responders to these peptides and correlation between the frequency of the tetramer-binding CD8+ T cells and that of IFN-γ-producing cells in ICS assay were performed by using two-tailed Pearson's correlation test. All data were analyzed using GraphPad Prism 8.

### Reporting summary

Further information on research design is available in the Nature Research Reporting Summary linked to this article.

## Data availability

All data are available in the main article in the Supplemental Information or available from the authors upon reasonable requests, as are unique reagents used in this Article. The raw numbers for charts and graphs are available in the Source Data file whenever possible. Source data are provided with this paper.

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

## Acknowledgements

We thank for all volunteers who contributed to this study and Drs. M. Baba and Y. Ariumi (Kumamoto University) for collection of peripheral blood samples.

## Author contributions

M.T., H.G., N.K., and T.C. designed the study. N.K., Y.Z., T.C., and H.N. performed the experiments. N.K., Y.Z., T.C., and M.T. analyzed the data. M.T., H.G., S.O., N.K., T.C., and Y.Z. wrote the manuscript.

## Competing interests

The authors declare no competing interests.
