## [Peer Review File · Nature Communications]

REVIEWER COMMENTS

Reviewer #1 (Remarks to the Author):

In this manuscript by Nazomi Kuse et. al., the authors have evaluated CD8+ T cells in 17 individuals vaccinated with SARS-CoV-2 BNT162b2 mRNA vaccine. The authors have focused on HLA A24:02 specific CD8 T cells.

Vaccine induced CD8 T cells were identified using PBMCs cultured with peptide-pools (previously published 16 CD8 T cells epitopes from spike protein) and IL-2 for 14 days and measuring cytokines release in expanded T cells upon incubation with peptide-loaded C1R-A24 cells. The functional characterization of epitope-specific CD8 T cells was performed up to 32 weeks after two vaccination doses. The authors further interrogated antigen-sensitivity and stability of pMHC complexes in relation to their immunodominance and cross-reactivity with the epitopes derived from four other human coronaviruses (H-CoVs).

Novel aspect of the manuscript is the identification of two vaccine-specific immunodominant HLA-A24 epitopes in this patient group that could also be population specific due to prevalence of the HLA-A24 (most prevalent) in the Asian population. These two epitopes have been identified previously in BNT162b2 vaccinated individuals and in SARS-CoV-2 infection, thus identifies their potential significance in disease protection. The authors further identify functional persistence of one of T cells specific to S-QI9 specific epitopes using cytokine analysis. Importantly, the authors provide valuable information on the stability and potential TCR sensitivity of the immunodominant and non-immunodominant/non-immunogenic peptides using TAP deficient RMA-S cells and correlated their stability with similar peptides from H-CoVs.

Specific points:

1. Epitope-specific T cell activation was measured on expanded PBMCs. Although, expanded bulk T cells provides a higher frequency of T cells for experimental evaluation, they do not provide complete information for each of the peptides included in the expansion. This is due to initial frequency of specific T cells. For example, a higher initial frequency of a peptide-specific T cells will be preferentially expanded limiting proliferation of remaining epitope-specific T cells. Specifically, in this manuscript, a pool of 5-6 peptides was used for CD8 T cell expansion, and we do not know the initial frequency for each of the peptide-specific T cells. I suggest:
 - a. Direct ex-vivo measurement (before expansion) of PBMCs stimulated with peptides to determine initial frequency of antigen-specific T cells.
 - b. Provide details of peptides used in each peptide-pool. Were the two immunodominant epitopes in the same peptide-pool?
 - c. How many individuals detected with two or more of the A24 responses? How does immunodominance change in those individuals? How many of the samples were identified with more than one immunodominant epitope-specific T cells? This information is important to identify and compare the immunodominance of HLA-specific epitopes.
2. The two immunodominant epitopes have been previously identified in SARS-CoV-2 unexposed individuals. The authors have also compared sequence similarity with other H-CoVs derived peptides. A very relevant question is whether pre-existing T cells (as identified previously) contribute to the vaccine mediated CD8 T cell response or these T cells responses are completely de novo. In figure 1D, a higher initial frequency (at 0 week) of IFN-g producing T cell is visible for QI9. This could indicate pre-existing T cells. It's important in the context of this manuscript to evaluate the two immunodominant epitopes in correlation to pre-existing frequency and subsequent vaccine induced immunodominance. How does the immune response to vaccine correlate with the pre-existing frequency?
3. Fig. 2: Impact of peptide concentration on sensitivity, is it due to higher expanded frequency of S-NF9 and S-QI9 compared to other peptides? Please compare peptide concentration either with same amount of T cells for each specificity or as a fraction of activated T cells.
4. Fig. 4, authors claim to have a correlation between stability of pMHC and frequency of vaccine responders, this is not completely true since the strongest binder in RMA-S cell assay is S-RF10 peptide-specific T cells were identified only in one patient.
5. Please provide details or reference of spike-specific antibody measurements.
6. Methods: Please include details on how many PBMCs were used in T cell expansion experiments.

Reviewer #2 (Remarks to the Author):

Kuse and co-authors investigate the SARS-CoV-2-specific CD8+T cell response related to 16 epitopes restricted to HLA-A*24:02 highly frequent worldwide and specifically in the Asian population. They found that less than 50% of those are recognized in 17 recipients of BNT162b2 vaccinees carrying the allele and specifically 3 of them were found to be the most immunodominant. While lacking a comprehensive approach in defining T cell responses and HLA repertoire, this study provides in-depth immune-mechanism using a single HLA example to confirm previous studies related to T cell memory-specific Vaccine longevity. This study supports immunodominance of specific HLA-A*24:02 epitopes previously reported in the literature, putting them in the context of HLA binding affinity and similarities with Common Cold Coronaviruses.

Point to address:

1-This study compares previous data collected in natural infection with vaccination and reaches conclusions about immunodominance. Considering the difference in antigen exposure between natural infection and vaccination, the points of the co-authors would be strengthened by comparing their results in naturally infected individuals.

2-The approach of in-vitro expansion and stimulation is powerful in characterizing in-depth T cell lines functionality and immunodominance, defined as the capability of specific T cell clones to expand more efficiently than others. However, this approach does not provide information related to response frequency upon antigen/epitope encounter. The comparison of ex-vivo and in-vitro findings in the same individuals would help the authors make this point, critical to perform an accurate comparison with the epitopes defined by the AIM assay. Additionally, this comparison will also help provide information on functionality in an ex-vivo system that was not co-cultured for 14 days in the presence of IL-2. In that regard, the following discussion statement is problematic: "Furthermore, cultured S-QI9- and 309 S-NF9-specific T cells produced IL-2 after stimulation with peptides in vitro, implying that these cultured cells still possessed characteristics of effector memory T cells." IL-2 addition does promote growth, proliferation, and differentiation from naive to effector memory T cells. In fact, it is not recommended to perform immuno-phenotyping on expanded T cells as the 14 days culture with IL-2 can cause also de-novo activation and phenotype alterations.

3-Several statements provided by the authors are approximative and reflect an outdated knowledge of the current SARS-CoV-2 literature that should be changed, even if discussing this will more evidently reduce the novelty of the study. Here are some examples:

a- "Indeed, it has been reported that antibody titers became lower at 4–6 months after vaccination, and the effectiveness of antibodies against the Delta variant was reduced."

With the recent insurgence of Omicron, it would be advisable to update the literature references accordingly.

b- "Although a large number of T cell-specific epitopes have been reported (<https://www.iedb.org>), these epitopes have not been well characterized".

Over 60 independent studies reported epitopes, if authors wish to make this point they should go more in-depth in this statement as the variety of techniques and approaches used for epitope identification may go against the Authors' statement.

c- "It remains controversial as to how long effective SARS-CoV-2 immunity persists in vaccinated individuals. Immunity has been evaluated based on levels of antibodies against SARS-CoV-2 spike antigens or neutralizing antibodies."

There are currently five or more studies addressing T cell response in SARS-CoV-2 Vaccination using multiple vaccine platforms.

Reviewer #3 (Remarks to the Author):

In the manuscript by Kuse et al the authors describe the induction of CD8+ T cell responses restricted by HLA-A24:02 against SARS-CoV-2 after mRNA vaccination. They in particular identify 2 dominant epitopes in several individuals after vaccination for a prolonged period (S-QI9 and S-NF9). The S-QI9 specific T cells were shown to elicit cross-reactivity against variant peptides from seasonal corona-viruses. The findings are of importance and warrant publication to the wide scientific community.

Some smaller points of improvements:

Make it possible to individually follow the different subjects through the figures and within the figures by using subject-specific numbers or symbols.

Fig2. Not fully clear if the points in the lines represent average, median or representative experiments.

Fig 3. Not super clear figure legend with regards to the calculation. Please revise.

Fig 4. Se first point with regards to symbols.

Discussion: be a little bit more speculative of how your findings can be used with regards to vaccination and cross-reactivity with seasonal corona-strains.

Best regards Michael Uhlin

Responses to comments from Reviewer #1:

In this manuscript by Nazomi Kuse et. al., the authors have evaluated CD8+ T cells in 17 individuals vaccinated with SARS-CoV-2 BNT162b2 mRNA vaccine. The authors have focused on HLA A24:02 specific CD8 T cells.

Vaccine induced CD8 T cells were identified using PBMCs cultured with peptide-pools (previously published 16 CD8 T cells epitopes from spike protein) and IL-2 for 14 days and measuring cytokines release in expanded T cells upon incubation with peptide-loaded C1R-A24 cells. The functional characterization of epitope-specific CD8 T cells was performed up to 32 weeks after two vaccination doses. The authors further interrogated antigen-sensitivity and stability of pMHC complexes in relation to their immunodominance and cross-reactivity with the epitopes derived from four other human coronaviruses (H-CoVs).

Novel aspect of the manuscript is the identification of two vaccine-specific immunodominant HLA-A24 epitopes in this patient group that could also be population specific due to prevalence of the HLA-A24 (most prevalent) in the Asian population. These two epitopes have been identified previously in BNT162b2 vaccinated individuals and in SARS-CoV-2 infection, thus identifies their potential significance in disease protection. The authors further identify functional persistence of one of T cells specific to S-QI9 specific epitopes using cytokine analysis. Importantly, the authors provide valuable information on the stability and potential TCR sensitivity of the immunodominant and non-immunodominant/non-immunogenic peptides using TAP deficient RMA-S cells and correlated their stability with similar peptides from H-CoVs.

Specific points:

1. Epitope-specific T cell activation was measured on expanded PBMCs. Although, expanded bulk T cells provides a higher frequency of T cells for experimental evaluation, they do not provide complete information for each of the peptides included in the expansion. This is due to initial frequency of specific T cells. For example, a higher initial frequency of a peptide-specific T cells will be preferentially expanded limiting proliferation of remaining epitope-specific T cells. Specifically, in this manuscript, a pool of 5-6 peptides was used for CD8 T cell expansion, and we do not know the initial frequency for each of the peptide-specific T cells. I suggest:

1-a. Direct ex-vivo measurement (before expansion) of PBMCs stimulated with peptides to determine initial frequency of antigen-specific T cells.

Reply: Previous studies showed that the frequency of T cells specific for each COVID-19 epitope in *ex vivo* PBMCs is very low (>0.01%–0.001%), suggesting that it is difficult to precisely determine the initial frequency of the peptide-specific T cells by an *ex vivo* ICS assay or an *ex vivo* ELISpot assay because of the high background of non-specific responses. In the present study, we used HLA-A*24:02-tetramers to precisely detect SARS-CoV-2 epitope-specific, HLA-A*24:02-restricted T cells. We constructed HLA-A*24:02-tetramers with two immunodominant epitopes (S-QI9 and S-NF9) and stained *ex vivo* PBMCs from 15 individuals (PBMCs from 2 of 17 individuals were not available for this assay because of the limited number of PBMC samples) at an earlier stage post-vaccination and 14 individuals at a later stage post-vaccination. The results with the tetramer-binding T cells in *ex vivo* PBMCs are shown in Fig. 2 (at an earlier stage post-vaccination) and in Fig. 6B and 6C (at a later stage post-vaccination), as well as in Fig. 5B–5D (for convalescent individuals previously infected with SARS-CoV-2). We present these results in lines 12-27, page 6, lines 6-17, page 8, and lines 6-13, page 9 as well as materials and methods of tetramer construction and tetramer staining in materials and methods section.

1-b. Provide details of peptides used in each peptide-pool. Were the two immunodominant epitopes in the same peptide-pool?

Reply: We added information regarding the peptides used in each peptide pool to Table S2. Immunodominant epitopes S-QI9 and S-NF9 are not included in the same peptide pool.

1-c. How many individuals detected with two or more of the A24 responses? How does immunodominance change in those individuals? How many of the samples were identified with more than one immunodominant epitope-specific T cells? This information is important to identify and compare the immunodominance of HLA-specific epitopes.

Reply: Two or more of the HLA-A*24:02-restricted responses were found in 11 of 17 individuals tested (64.7%) whereas T cell responses to S-QI9 and S-NF9 were detected in 13 and 10 individuals, respectively. At least one of two immunodominant responses to S-QI9 and S-NF9 were detected in 15 individuals (88.2%), indicating that these two epitopes are immunodominant. The numbers of responders to HLA-

A*24:02-restricted epitopes and those to HLA-A*24:02-restricted immunodominant epitopes are shown in Fig. 1C and 1D, respectively. As a response to S-VYF10 was only detected in 3 out of 17 individuals (17.6%) and the magnitude of T cell expansion was low (less than 5%), we concluded that S-VYF10 is a subdominant epitope. These findings are mentioned in lines 20-24, page5.

2: The two immunodominant epitopes have been previously identified in SARS-CoV-2 unexposed individuals. The authors have also compared sequence similarity with other H-CoVs derived peptides. A very relevant question is whether pre-existing T cells (as identified previously) contribute to the vaccine mediated CD8 T cell response or these T cells responses are completely de novo. In figure 1D, a higher initial frequency (at 0 week) of IFN-g producing T cell is visible for QI9. This could indicate pre-existing T cells. It's important in the context of this manuscript to evaluate the two immunodominant epitopes in correlation to pre-existing frequency and subsequent vaccine induced immunodominance. How does the immune response to vaccine correlate with the pre-existing frequency?

Reply: Thank you for your comments on pre-existing T cells. As you mentioned, the findings from the ICS assay of cultured T cells demonstrated that one (V12) out of three individuals had a low frequency of S-QI9-specific T cells at prior vaccination (Fig. 1F). However, additional analysis using tetramers did not detect S-QI9-specific T cells in *ex vivo* PBMCs collected from the same individual at the same time point (time 0) (Fig. 2E). A similar finding was obtained prior to vaccination in another individual (V10), and in this case, the tetramer binding assay demonstrated S-QI9-specific T cells in *ex vivo* PBMCs whereas the T cells were not detected in the ICS assay of cultured T cells (Fig. 1F and 2E). These findings suggested that pre-existing S-QI9-specific T cells may exist at a very low frequency in these two individuals. As S-QI9-specific T cells were detected at a high frequency in V12 but at a low frequency in V10 after vaccination, it is speculated that vaccine-induced S-QI9-specific T cells may result from the expansion of pre-existing T cells. As we only had pre-vaccination samples from three individuals, it is difficult in the current study to evaluate the correlation between pre-existing T cell frequency and subsequent vaccine-induced immunodominance. We mention this issue in the revised Results and Discussion sections (lines 8-24, page 13, Discussion section).

3. Fig. 2: Impact of peptide concentration on sensitivity, is it due to higher expanded

frequency of S-NF9 and S-QI9 compared to other peptides? Please compare peptide concentration either with same amount of T cells for each specificity or as a fraction of activated T cells.

Reply: Thank you for your suggestion. We used cultured bulk T cells specific for each epitope in this analysis. To compare peptide concentrations conferring approximately 50% of the maximum T cell response in T cells specific for each peptide, we changed the Y-axis of Figure 3 (Fig. 2 in the original manuscript) from “the frequency of specific T cells among CD8+ T cells” to “relative frequency of the specific T cells compared with their maximum responses at 100 nM (NF9 and QI9 immunodominant epitopes) or 1,000 nM (other epitopes)”. The results showed that NF9-specific and QI9-specific T cells have higher sensitivity than other T cells (0.1–1 nM versus 10–1,000 nM). This is mentioned in lines 3-8, page 7.

4. **Fig. 4**, authors claim to have a correlation between stability of pMHC and frequency of vaccine responders, this is not completely true since the strongest binder in RMA-S cell assay is S-RF10 peptide-specific T cells were identified only in one patient.

Reply: We agree with this comment. We revised the sentence explaining the data for Fig. 4B (lines 14-15, page 7).

5. Please provide details or reference of spike-specific antibody measurements.

Reply: We present the antibody data for the vaccinated individuals in Table S1.

6. Methods: Please include details on how many PBMCs were used in T cell expansion experiments.

Reply: For each peptide pool in the T cell expansion experiments, 10^6 PBMCs were used. This information has been added to the Materials and Methods section (line 18, page 15).

Responses to comments from Reviewer #2:

Kuse and co-authors investigate the SARS-CoV-2-specific CD8+T cell response related

to 16 epitopes restricted to HLA-A*24:02 highly frequent worldwide and specifically in the Asian population. They found that less than 50% of those are recognized in 17 recipients of BNT162b2 vaccinees carrying the allele and specifically 3 of them were found to be the most immunodominant. While lacking a comprehensive approach in defining T cell responses and HLA repertoire, this study provides in-depth immunomechanism using a single HLA example to confirm previous studies related to T cell memory-specific Vaccine longevity. This study supports immunodominance of specific HLA-A*24:02 epitopes previously reported in the literature, putting them in the context of HLA binding affinity and similarities with Common Cold Coronaviruses.

Point to address:

1-This study compares previous data collected in natural infection with vaccination and reaches conclusions about immunodominance. Considering the difference in antigen exposure between natural infection and vaccination, the points of the co-authors would be strengthened by comparing their results in naturally infected individuals.

Reply: As this reviewer suggested, we analyzed PBMC samples from 10 HLA-A*24:02⁺ convalescent individuals previously infected with SARS-CoV-2 that had not been vaccinated by an ICS assay of cultured T cells stimulated with the HLA-A*24:02 epitope peptides (Fig. 5A) and a binding assay of the *ex vivo* PBMCs with the HLA-A*24:02 tetramer of two immunodominant epitopes (Fig. 5B). Two immunodominant-specific T cells were predominantly found in the vaccinated and convalescent individuals, although the frequency of these immunodominant epitope-specific T cells was lower in the convalescent individuals than in the vaccinated individuals (Fig. 5 versus Fig. 1). Regarding long-term memory T cells specific for two immunodominant epitopes, these T cells were elicited and had the capacity to proliferate *in vitro* in the vaccinated individuals and the convalescent individuals within 30 weeks of diagnosis of the infection (Fig. 5C and 5D). By contrast, these immunodominant memory T cells may lose their ability to proliferate more than 30 weeks after diagnosis (Fig. 5C and 5D). These findings are presented in the Results section entitled “HLA-A*24:02-restricted CD8⁺ T cells elicited in convalescent individuals previously infected with SARS-CoV-2” and the Discussion section (line 24, page 11- line 2, page 12).

2-a: The approach of in-vitro expansion and stimulation is powerful in characterizing in-depth T cell lines functionality and immunodominance, defined as the capability of specific T cell clones to expand more efficiently than others. However, this approach does not provide information related to response frequency upon antigen/epitope encounter. The comparison of ex-vivo and in-vitro findings in the same individuals would help the authors make this point, critical to perform an accurate comparison with the epitopes defined by the AIM assay.

Reply: As we responded to the first comment from Reviewer #1, we used HLA-A*24:02-tetramers to detect T cells specific for immunodominant epitopes, S-QI9 and S-NF9, in *ex vivo* PBMCs. The results of the tetramer binding assay are shown in Fig. 2 (at an early stage) and Fig. 6B and 6C (at a later stage). In addition, we analyzed the correlation between the frequency of T cells specific for S-QI9 and S-NF9 in *ex vivo* PBMCs and that of expanded T cells *in vitro*. Strong correlations between these two frequencies were found at both the early (Fig. 2C) and later phases (Fig. 6C). These findings indicated that two immunodominant T cells were frequently elicited and maintained, with the capacity to proliferate by epitope peptide stimulation in vaccinated individuals. This is mentioned in lines 12-17, page 6, lines 6-17, page 8, and lines 6-13, page 9. Please also see our reply to the first comment from Reviewer #1.

2-b. Additionally, this comparison will also help provide information on functionality in an ex-vivo system that was not co-cultured for 14 days in the presence of IL-2. In that regard, the following discussion statement is problematic: "Furthermore, cultured S-QI9- and 309 S-NF9-specific T cells produced IL-2 after stimulation with peptides in vitro, implying that these cultured cells still possessed characteristics of effector memory T cells." IL-2 addition does promote growth, proliferation, and differentiation from naive to effector memory T cells. In fact, it is not recommended to perform immunophenotyping on expanded T cells as the 14 days culture with IL-2 can cause also de-novo activation and phenotype alterations.

Reply: Thank you for your comments. As this reviewer suggested, we deleted these data (Figure 4 in the original manuscript) in our revised manuscript.

3-Several statements provided by the authors are approximative and reflect an outdated knowledge of the current SARS-CoV-2 literature that should be changed, even if

discussing this will more evidently reduce the novelty of the study. Here are some examples:

3-a- "Indeed, it has been reported that antibody titers became lower at 4–6 months after vaccination, and the effectiveness of antibodies against the Delta variant was reduced." With the recent resurgence of Omicron, it would be advisable to update the literature references accordingly.

Reply: Thank you for your comments. We have revised this section, as suggested (lines 10 - 12, page 3).

3-b "Although a large number of T cell-specific epitopes have been reported (<https://www.iedb.org>), these epitopes have not been well characterized".

Over 60 independent studies reported epitopes, if authors wish to make this point they should go more in-depth in this statement as the variety of techniques and approaches used for epitope identification may go against the Authors' statement.

Reply: Thank you for your comment. We revised this section, as suggested (lines 13 -14 and lines 18-24, page 3).

3-c "It remains controversial as to how long effective SARS-CoV-2 immunity persists in vaccinated individuals. Immunity has been evaluated based on levels of antibodies against SARS-CoV-2 spike antigens or neutralizing antibodies."

There are currently five or more studies addressing T cell response in SARS-CoV-2 Vaccination using multiple vaccine platforms.

Reply: Thank you for your comment. We revised this section, as suggested (line 28,page 3-line 2, page 4 and lines 4-5, page 4), and added several references (Reference # 41-47).

Responses to comments from Reviewer #3.

In the manuscript by Kuse et al the authors describe the induction of CD8+ T cell responses restricted by HLA-A24:02 against SARS-CoV-2 after mRNA vaccination. They in particular identify 2 dominant epitopes in several individuals after vaccination for a prolonged period (S-QI9 and S-NF9). The S-QI9 specific T cells were shown to

elicit cross-reactivity against variant peptides from seasonal corona-viruses. The findings are of importance and warrant publication to the wide scientific community.

Reply: Thank you for your positive comments on our study.

Some smaller points of improvements:

1. Make it possible to individually follow the different subjects through the figures and within the figures by using subject-specific numbers or symbols.

Reply: As suggested, we discriminated the vaccinated individuals according to different symbols and colors in Fig. 1A, 1E, 2B, 2D, 5A–D, 6A, and 6B. Symbols and colors were used consistently for each individual throughout the figures.

2. Fig2. Not fully clear if the points in the lines represent average, median or representative experiments.

Reply: We present the mean \pm SD (N=3) values for the T cells specific for the six epitopes in the revised version of this figure (Fig. 3). However, as we did not have a sufficient number of S-VF12-specific T cells for triplicate assays, the data for these T cells only represent a single assay. As Reviewer #1 suggested, we revised the X-axis of the figure (please see our response to comment 3 from Reviewer #1).

3. Fig 3. Not super clear figure legend with regards to the calculation. Please revise.

Reply: We revised the figure legends for the peptide binding data. The details of this assay have been added to the Materials and Methods section entitled “HLA stabilization assay using RMA-S-A2402 cells”.

4. Fig 4. Se first point with regards to symbols.

Reply: We revised this figure (Fig. 6A in the revised version) as we mentioned in response to comment 1 from this reviewer.

5. Discussion: be a little bit more speculative of how your findings can be used with regards to vaccination and cross-reactivity with seasonal corona-strains.

Reply: Thank you for your comment. We have expanded the discussion of pre-existing T cells cross-recognizing the S-QI9 epitope (paragraph 4 of the Discussion section, pages 12–13) and cross-reactivity with seasonal coronaviruses (lines 5 - 24, page 13).

REVIEWERS' COMMENTS

Reviewer #1 (Remarks to the Author):

The authors have resolved all my comments and made relevant changes in the manuscript. I do not have any additional comments.

Reviewer #2 (Remarks to the Author):

The reviewer have addressed all the points raised by the reviewers. The inclusion of ex-vivo data using tetramer staining and the comparison with natural infection complement and strengthen the points raised by the author. I do believe that this manuscript has considerably improved and worth publication in Nature Communication.

Reviewer #3 (Remarks to the Author):

The authors have addressed the raised questions and concerns.